# Long-term cognitive effects of menopausal hormone therapy: Findings from the KEEPS Continuation Study

**Carey E. Gleason**[1,2☯*], **N. Maritza Dowling**[3☯], **Firat Kara**[4☯], **Taryn T. James**[1], **Hector Salazar**[5], **Carola A. Ferrer Simo**[1], **Sherman M. Harman**[6], **JoAnn E. Manson**[7], **Dustin B. Hammers**[8], **Frederick N. Naftolin**[9], **Lubna Pal**[10], **Virginia M. Miller**[11], **Marcelle I. Cedars**[12], **Rogerio A. Lobo**[13], **Michael Malek-Ahmadi**[14], **Kejal Kantarci**[4]

1 Department of Medicine, University of Wisconsin, Madison, Wisconsin, United States of America, 2 Geriatric Research, Education and Clinical Center (GRECC), William S. Middleton Memorial VA Hospital, Madison, Wisconsin, United States of America, 3 Department of Acute & Chronic Care, George Washington University, Washington, DC, United States of America, 4 Department of Radiology, Mayo Clinic, Rochester, Minnesota, United States of America, 5 Department of Health and Community Systems, University of Pittsburgh School of Nursing, Pittsburgh, Pennsylvania, United States of America, 6 Phoenix VA Health Care System, Phoenix, Arizona, United States of America, 7 Department of Medicine, Brigham and Women's Hospital, Harvard Medical School, Boston, Massachusetts, United States of America, 8 Department of Neurology, Indiana University School of Medicine, Indianapolis, Indiana, United States of America, 9 e-Bio Corp., New York, New York State, United States of America, 10 Department of Obstetrics, Gynecology and Reproductive Sciences, Yale University, New Haven, Connecticut, United States of America, 11 Department of Surgery, Mayo Clinic, Rochester, Minnesota, United States of America, 12 Department of Obstetrics and Gynecology, University of California, San Francisco, California, United States of America, 13 Department of Obstetrics and Gynecology, Columbia University, New York, New York State, United States of America, 14 Banner Alzheimer Institute Phoenix, Arizona, United States of America

☯ These authors contributed equally to this work.
* ceg@medicine.wisc.edu

**Data Availability Statement:** The data is stored in the Medidata repository. Qualified academic and industry researchers can request data, from the

## Abstract

### Background

Findings from Kronos Early Estrogen Prevention Study (KEEPS)-Cog trial suggested no cognitive benefit or harm after 48 months of menopausal hormone therapy (mHT) initiated within 3 years of final menstrual period. To clarify the long-term effects of mHT initiated in early postmenopause, the observational KEEPS Continuation Study reevaluated cognition, mood, and neuroimaging effects in participants enrolled in the KEEPS-Cog and its parent study the KEEPS approximately 10 years after trial completion. We hypothesized that women randomized to transdermal estradiol (tE2) during early postmenopause would show cognitive benefits, while oral conjugated equine estrogens (oCEE) would show no effect, compared to placebo over the 10 years following randomization in the KEEPS trial.

### Methods and findings

The KEEPS-Cog (2005–2008) was an ancillary study to the KEEPS (NCT00154180), in which participants were randomized into 3 groups: oCEE (Premarin, 0.45 mg/d), tE2 (Climara, 50 μg/d) both with micronized progesterone (Prometrium, 200 mg/d for 12 d/mo) or placebo pills and patch for 48 months. KEEPS Continuation (2017–2022), an observational,

Mayo Clinic Alzheimer's Disease Research Center. Once a request is submitted, the committee sends the indicated principal investigator an email confirming that the request was received and giving a timeline for committee review. The data can be requested by filling data request form using the following link. https://www.mayo.edu/research/centers-programs/alzheimers-disease-research-center/data-requests. For your questions, please email the Data Sharing Coordinator at mcsaadrcdatasharing@mayo.edu.

**Funding:** Financial disclosure: Financial disclosure: Authors also gratefully acknowledge the support of Yale Center for Clinical Investigation, and the support provided by CTSA Grant Number UL1 TR001863 from the National Center for Advancing Translational Science (NCATS), a component of the National Institutes of Health (NIH) to the Yale Center for Clinical Investigation (to LP); and support from the Brigham and Women's Hospital (BWH)/Harvard Medical School Clinical and Translational Science Award UL1 RR024139 from NCATS (to JEM), a component of NIH, to the BWH Center for Clinical Investigation; support from the National Institute on Aging Award (1RF1AG057547 to KK). Contents of this paper are solely the responsibility of the authors and do not necessarily represent the official view of NIH. The funders had no role in the study design, data collection, data analysis, data interpretation, or writing of the report.

**Competing interests:** K.K. served on the data safety monitoring board for Pfizer Inc. and Takeda Global Research & Development Center, Inc. She received research support from Avid. MM-A received consulting fees from the Biomedical Research Alliance of New York. Radiopharmaceuticals, Eli Lilly. She consults for Biogen.

**Abbreviations:** AAWM, auditory attention and working memory; BMI, body mass index; CAC, coronary artery calcium; CFI, comparative fit index; CHD, coronary heart disease; CIMT, carotid intima-media thicknesstest; CVD, cerebrovascular disease; ELITE, Early versus Late Intervention Trial with Estradiol; HT, hormone therapy; KEEPS, Kronos Early Estrogen Prevention Study; KIWI, Kinmen women-health investigation; LGM, latent growth model; mHT, menopausal hormone therapy; MPA, medroxyprogesterone acetate; oCEE, oral conjugated equine estrogens; PET, positron emission tomography; PI, principal investigator; SBP, systolic blood pressure; SLMF, speeded language and mental flexibility; SWAN, Study of Women Across the Nation; VAEF, visual attention and executive function; VLM, verbal

longitudinal cohort study of KEEPS clinical trial, involved recontacting KEEPS participants approximately 10 years after the completion of the 4-year clinical trial to attend in-person research visits. Seven of the original 9 sites participated in the KEEPS Continuation, resulting in 622 women of original 727 being invited to return for a visit, with 299 enrolling across the 7 sites. KEEPS Continuation participants repeated the original KEEPS-Cog test battery which was analyzed using 4 cognitive factor scores and a global cognitive score. Cognitive data from both KEEPS and KEEPS Continuation were available for 275 participants. Latent growth models (LGMs) assessed whether baseline cognition and cognitive changes during KEEPS predicted cognitive performance at follow-up, and whether mHT randomization modified these relationships, adjusting for covariates.

Similar health characteristics were observed at KEEPS randomization for KEEPS Continuation participants and nonparticipants (i.e., women not returning for the KEEPS Continuation). The LGM revealed significant associations between intercepts and slopes for cognitive performance across almost all domains, indicating that cognitive factor scores changed over time. Tests assessing the effects of mHT allocation on cognitive slopes during the KEEPS and across all years of follow-up including the KEEPS Continuation visit were all statistically nonsignificant.

The KEEPS Continuation study found no long-term cognitive effects of mHT, with baseline cognition and changes during KEEPS being the strongest predictors of later performance. Cross-sectional comparisons confirmed that participants assigned to mHT in KEEPS (oCEE and tE2 groups) performed similarly on cognitive measures to those randomized to placebo, approximately 10 years after completion of the randomized treatments. These findings suggest that mHT poses no long-term cognitive harm; conversely, it provides no cognitive benefit or protective effects against cognitive decline.

## Conclusions

In these KEEPS Continuation analyses, there were no long-term cognitive effects of short-term exposure to mHT started in early menopause versus placebo. These data provide reassurance about the long-term neurocognitive safety of mHT for symptom management in healthy, recently postmenopausal women, while also suggesting that mHT does not improve or preserve cognitive function in this population.

## Author summary

### Why was this study done?

Little is known about the long-term cognitive effects of short-term use of menopausal hormone therapies (mHT)—i.e., the use of mHT during the menopausal transition or in early postmenopause for symptoms of menopause, leaving women and their providers with concerns about long-term consequences of short-term mHT use. We invited women who participated in a study examining the cognitive effects of short-term mHT to return for re-evaluation approximately a decade after they were randomized to 4 years of treatment with one of 2 forms of mHT or a placebo. Importantly, the original study only enrolled women who were recently postmenopausal and at low cardiovascular risk. The

learning and memory; WHI, Women's Health Initiative; WHIMS, Women's Health Initiative Memory Study; WHIMSY, Women's Health Initiative Memory Study of Younger Women.

goals for the follow-up study were to examine the long-term cognitive effects of using mHT for a brief period shortly after menopause onset and to assess if these effects differed for the 2 forms of mHT.

## What did the researchers do and find?

- The observational Kronos Early Estrogen Prevention Study (KEEPS) Continuation explored cognitive effects of short-term (4 year) randomized assignment to mHT versus placebo, initiated within 3 years of menopause, after an average of 10 years following randomization in the original KEEPS trial.

- We tested whether long-term cognitive performance was influenced by prior exposure to mHT formulation (e.g., transdermal 17β-estradiol or oral conjugated equine estrogens), controlling for covariates using linear latent growth models.

- Among the women enrolled in the KEEPS Continuation, cognitive performance was not influenced by earlier exposure to either mHT formulation.

- Linear growth models showed strong associations between baseline cognition (intercept) and its change (slope) during KEEPS and the same measures in the KEEPS Continuation.

- KEEPS Continuation cross-sectional comparisons confirmed that both oral and transdermal mHT groups performed similarly to placebo on cognitive measures approximately 10 years after they were randomized to either mHT or placebo.

## What do these findings mean?

- There has been ongoing debate regarding the long-term effects of mHT. By conducting a longitudinal observational follow-up of a clinical cohort, we found no long-term cognitive benefit or harm associated with short-term mHT compared to placebo. This is a novel finding, as it is based on an observational cohort that extends the results of a placebo-controlled randomized clinical trial.

- Our findings suggest that short-term mHT exposure in recently postmenopausal women with low cardiovascular risk has no long-term impact on cognition. These results are significant for public health, offering reassurance to women who are considering mHT to manage menopausal symptoms. The data indicate no negative long-term effects on cognition for women with good cardiovascular health.

- On the other hand, these finding also confirmed that mHT was not associated with cognitive benefits nor does mHT prevent cognitive decline. Therefore, mHT should not be recommended as an intervention to preserve cognitive function in postmenopausal women.

- These findings highlight the need for further research to explore other potential long-term health outcomes associated with mHT, beyond cognitive performance. Future studies could focus on areas such as mood and Alzheimer's disease biomarkers.

- Although this study is based on robust follow-up data, there are some limitations. Not all original participants returned for follow-up, and those who did may have had

different health characteristics, such as higher baseline cognitive scores. Additionally, the study only examined the effects of mHT in healthy women with low cardiovascular risk, so the findings may not be generalizable to women with other health conditions or those who begin mHT later in life.

## Introduction

It is estimated that around three-quarters of women will experience symptoms linked to the menopausal transition and approximately one-quarter describe the symptoms as moderately to severely bothersome [1,2]. Common symptoms include vasomotor (hot flashes) and vaginal (vulvovaginal atrophy) symptoms, disturbed sleep, depressed mood, and cognitive difficulties [3,4]. The most effective treatment for these symptoms is menopausal hormone therapy (mHT) [5]. Still, women and their health care providers avoid using mHT based on concerns about its safety, as highlighted in a prominent lay audience publication [6].

Concern about the safety of mHT stems in part from the unexpected findings from the Women's Health Initiative (WHI) study and its ancillary Memory Study (WHIMS). Specifically, findings from the WHI indicated that treatment with oral conjugated equine estrogens (oCEE) plus medroxyprogesterone acetate (MPA) was associated with elevations in risk for coronary heart disease (CHD) and cerebrovascular disease (CVD) events in older women, who were more than 10 years from the onset of menopause—in addition to the known risk for breast cancer [7,8]. Directly related to cognition, the WHIMS examined the effects of oCEE with MPA and oCEE-alone administered to women aged 65 and older. Altogether, the WHIMS found that both formulations of mHT exhibited deleterious effects on global cognitive function and risk for incident cognitive impairment [9,10], with oCEE + MPA in particular demonstrating an association with risk of incident mild cognitive impairment and dementia [11].

Typical use of mHT would rarely involve starting therapy at ages above 65. Critics of the WHIMS design were quick to highlight how initiating hormone therapy (HT) in older women may have profoundly different brain effects than starting therapy around onset of menopause, e.g., Henderson and Brinton [12], suggesting a critical window for initiation of mHT—one close to menopause onset. Discussion about a critical window led to a shift in terminology, distinguishing HT from mHT—therapy timed to occur during or close to the menopausal transition.

Randomized controlled clinical trials including the Kronos Early Estrogen Prevention Study (KEEPS) [13], and its ancillary Cognitive and Affective study (KEEPS-Cog), examined mHT use proximate to a final menstrual period (early postmenopausal mHT), finding no evidence of harm to cognitive performance with short term use of 2 different forms of mHT, oCEE and transdermal estradiol (tE2) [14,15]. Equally importantly, mood benefits were found for women treated with the oCEE formulation compared to placebo [14].

We report here the primary cognitive findings of the KEEPS Continuation Study, the primary aims of which were to examine the long-term effects of exposure to mHT on cognitive aging and Alzheimer's disease pathology. In the KEEPS Continuation, participants from the original KEEPS study were re-evaluated approximately a decade after their randomization to 4 years of exposure to one of 2 forms of mHT (oCEE or tE2 with cyclical micronized progesterone) versus placebo. **Fig 1** describes the chronological relationship between the 3 study phases.

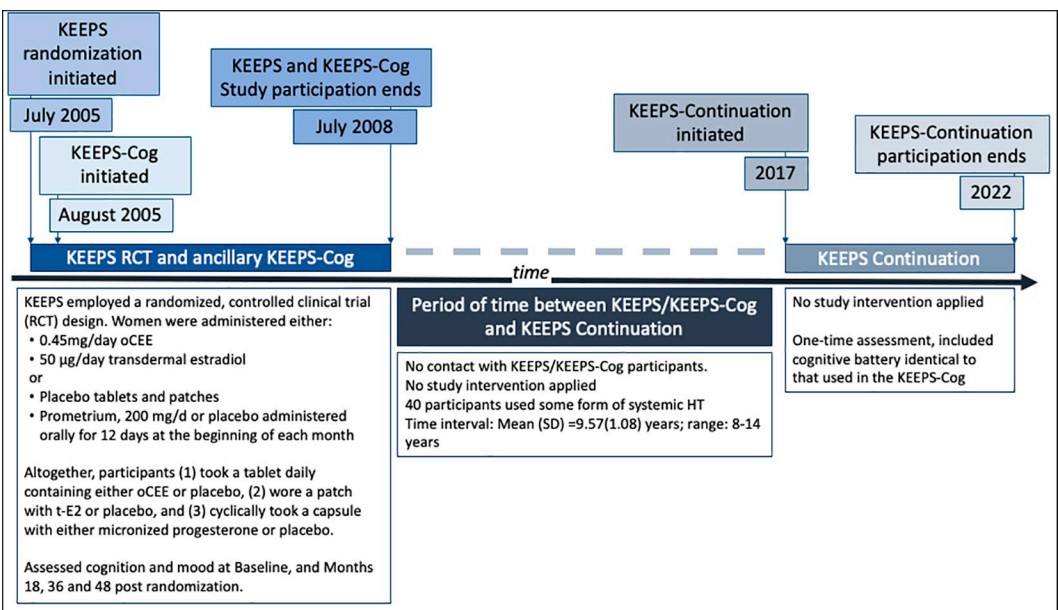

**Fig 1. Timeline for KEEPS, KEEPS-Cog, and KEEPS Continuation studies.** The KEEPS initiated randomization in July of 2005. In some instances, ancillary studies were started later. The KEEPS-Cognitive and Affective study was started in August of 2005. **KEEPS,** Kronos Early Estrogen Prevention Study. Parent study examining cardiovascular effects of early postmenopausal hormone therapy in women at low risk for cardiovascular disease. **KEEPS-Cog, KEEPS Cognitive and Affective study**, an ancillary study to the KEEPS, enrolled most but not all women enrolled in the KEEPS and studied cognitive and mood effects. **KEEPS Continuation,** Re-enrolled women in KEEPS to study long-term effects of mHT administered in early postmenopause.

We hypothesized that, compared to women treated with placebo, exposure to mHT would influence cognitive performance at long-term follow-up with the direction of influence differing depending upon the mHT formulation. Kantarci and colleagues [16] demonstrated that tE2 was associated with better preservation of prefrontal cortex volume compared to placebo, 7 years post-randomization in an ancillary study of KEEPS. The better preservation of the prefrontal cortex volume in tE2 group was associated with lower amyloid deposition. These neurobiological differences suggest that tE2 might confer cognitive advantages manifesting over a longer period, particularly in regions of the brain vulnerable to aging and Alzheimer's disease pathology. Building on these results, we hypothesized that tE2 would demonstrate cognitive benefits over placebo, and oCEE would show no difference from placebo.

## Methods

### KEEPS Continuation participants

The KEEPS Continuation Study conducted follow-up assessments of women previously enrolled in the parent KEEPS study, most of whom were also enrolled in the ancillary KEEPS-Cog (662 out of 727 or 91%). However, all women enrolled in the parent KEEPS (*n* = 727) were eligible for participation in the KEEPS Continuation. Outcomes assessed included cognition and mood, and neuroimaging for Alzheimer's disease proteinopathies; analyses presented here are limited to the cognitive outcomes.

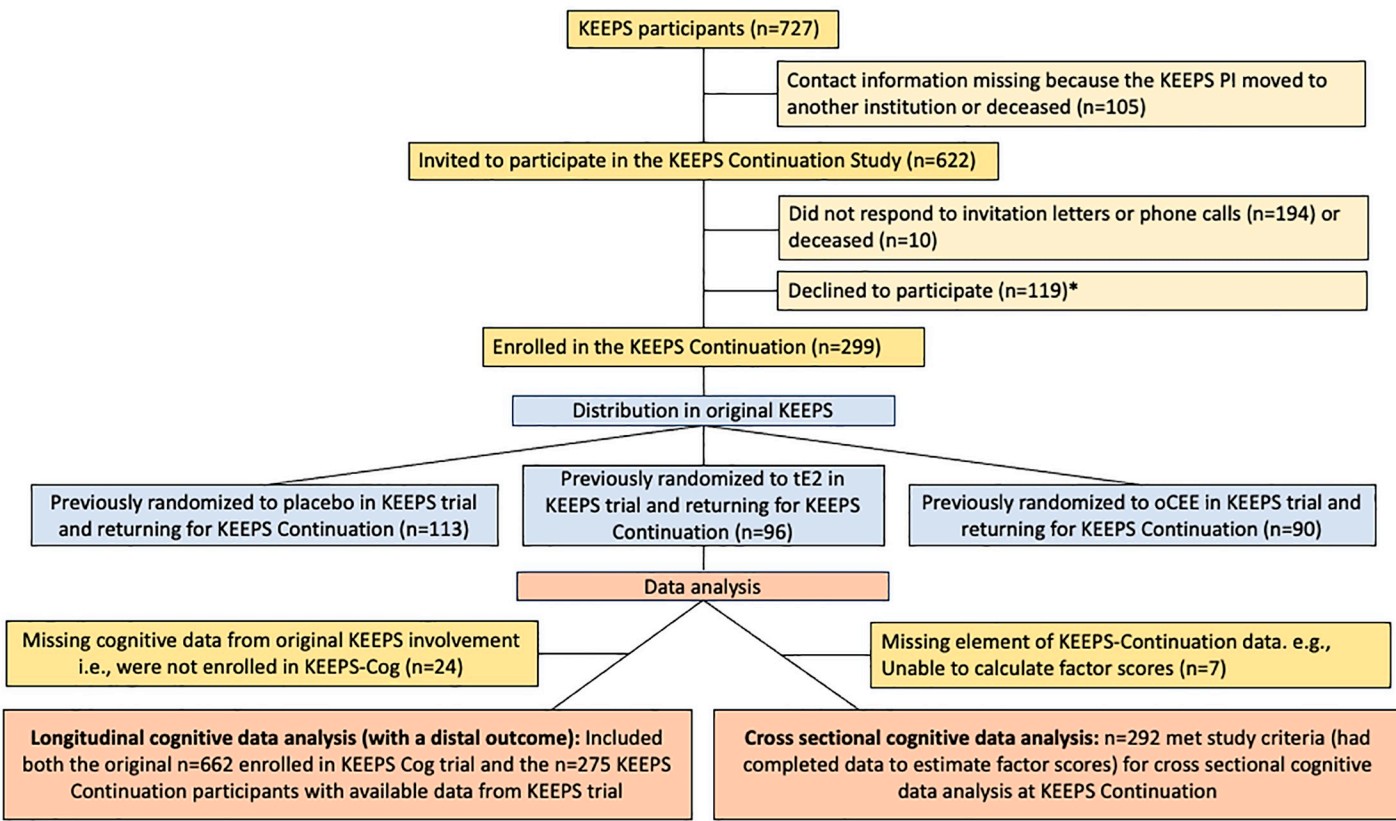

**Fig 2. Study flowchart.** *COVID-19-related concerns, inconvenient to travel, scheduling conflicts, or moved. KEEPS, Kronos Early Estrogen Prevention Study; oCEE, oral conjugated equine estrogens; PI, principal investigator; tE2, transdermal 17β-estradiol.

Of the 727 postmenopausal participants of KEEPS interventions, valid contact information was available for 622 (86%), all of whom were invited to participate in the KEEPS Continuation Study. Of the 622, 194 did not respond to the invitation, 10 were deceased, and 119 declined to participate. Overall, KEEPS Continuation enrolled 299 KEEPS trial participants at 7 sites (Albert Einstein College of Medicine-Montefiore, Banner Alzheimer's Institute, Brigham and Women's Hospital, Columbia University, Mayo Clinic, University of California San Francisco, University of Utah, and Yale University). Of the 299 KEEPS Continuation participants, 275 (overall, 92%; oCEE, 31%; tE2, 33%; placebo, 36%) had cognitive data available both from KEEPS and KEEPS Continuation (**Fig 2**). For the 387 KEEPS-Cog women who either declined to participate in KEEPS Continuation or who could not be located, cognitive data were obtained from the original KEEPS database. These women were considered a nonparticipant group in the current analysis of cognitive outcomes.

## Enrollment sites and ethics approvals

Seven of the original 9 sites participated in the KEEPS Continuation. One site's original contact PI was deceased (University of Washington). No new contact PI was available at another site (Albert Einstein College of Medicine-Montefiore), but these participants were evaluated at the Columbia University Site. The parent KEEPS PI (SMH) contacted participants enrolled at the Kronos Longevity Research Institute site, transferring interested participants to a contact PI at the Banner Institute. All other sites from the original KEEPS remained involved, serving

as primary contact for participants. Institutional Review Boards (IRB) at the 6 enrollment sites and the University of Wisconsin, Madison reviewed and approved the research protocol. The original KEEPS study's clinical trial registration number is NCT00154180. The KEEPS Continuation was an observational study and did not meet the criteria for registration with ClinicalTrials.gov. This study is reported as per the Strengthening the Reporting of Observational Studies in Epidemiology (STROBE) guideline (**S1 STROBE Checklist**).

## KEEPS and KEEPS Continuation

KEEPS employed a randomized, controlled clinical trial design wherein women were administered either 0.45 mg/day oCEE, 50 μg/day tE2, or placebo tablets and patches. All participants in KEEPS had an intact uterus. Therefore, micronized progesterone (Prometrium, 200 mg/d) was given orally for 12 days at the beginning of each month to mHT groups for endometrial protection [17]. Altogether, participants (1) took a tablet daily containing either oCEE or placebo; (2) wore a patch with tE2 or placebo; and (3) cyclically took a capsule with either micronized progesterone or placebo. Further details regarding study participants are available in publications describing the primary findings [14,17]. Enrollment in the original KEEPS occurred between July 2005 and June 2008. The women who participated KEEPS were with low CVD risk. Women were excluded if they had a history of clinically defined CVD, including myocardial infarction, angina, congestive heart failure, stroke, transient ischemic attack, or thromboembolic disease. Additional exclusion criteria included uncontrolled hypertension (systolic BP > 150 mm Hg or diastolic BP > 95 mm Hg), smoking more than 10 cigarettes daily, a body mass index (BMI) >35 kg/m2, diabetes (fasting glucose >126 mg/dL), dyslipidemia (total cholesterol >240 mg/dL), or a coronary artery calcium (CAC) score of 50 Agatston units or greater [17]. KEEPS rationale for focusing on a low CVD risk group study was to minimize confounding factors. At the time the study was designed, the Women's Health Initiative findings were being discussed widely and its study designed critiqued for their inclusion of women whose CVD risk was considered elevated [18]. The KEEPS investigators hypothesized that mHT would prevent incident CVD in women who were free of CVD at baseline. The primary outcomes of interest were carotid intima-media thicknesstest (CIMT) and CAC scores [13]. Women who had participated the original KEEPS trial were recontacted for the KEEPS Continuation Study, even if they did not participate in the KEEPS-Cog ancillary study or have baseline cognitive assessments. KEEPS Continuation enrollment occurred between May 2019 and June 2022. No medications or non-pharmaceutical interventions were administered in the KEEPS Continuation Study. Women were contacted for re-enrollment through their original enrollment site and invited to return for observational data collection visits. Data collected included medical history, cognitive and mood data collection, biometric examination, magnetic resonance imaging of the brain and positron emission tomography (PET) measurement of brain amyloid and tau (amyloid and tau PET). Comparison of the cardiometabolic status of women enrolled in the KEEPS Continuation and those in the original KEEPS were published previously [19]. Data collected from MRI and PET will be the focus of a future publication. This report focuses on the cognitive outcomes.

## Approach for recontacting

All participants enrolled in the original KEEPS were eligible for the KEEPS Continuation, not just those enrolled in the ancillary KEEPS-Cog. Thus, following IRB approval, all women enrolled in the original KEEPS were first sent informational letters from personnel at their original enrollment site, inviting them to enroll in the KEEPS Continuation Study. Follow-up telephone communication was attempted for those who did not respond to letters. If letters

were returned due to a change of address, staff conducted brief on-line searches for new address or contact information. Participants from the original KEEPS, who were enrolled at the Albert Einstein College of Medicine/Montefiore Medical Center site in New York, were approached for KEEPS Continuation enrollment at the Columbia University site in New York. Women enrolled at the Albert Einstein College of Medicine-Montefiore site were sent letters from a staff member involved in the original study, inviting them to contact the Columbia University site for enrollment into the KEEPS Continuation study. Site principal investigators (PIs) and their research staff had contact with the participants enrolled at their site. In most instances, the site PIs was the same person who led data collection for the original KEEPS. Each site obtained IRB approval to re-contact the participants enrolled at their site. Participants signed site-specific consent forms reviewing all study procedures and the sharing of data with the Mayo Clinic and the University of Wisconsin, Madison, allowing all sites' data to be collated for analyses. Kejal Kantarci served as site PI for the Mayo Clinic in place of Virginia Miller, the original KEEPS Mayo Clinic site PI. Outside of this subset of participants, the study main PIs (Carey E. Gleason and Kejal Kantarci) did not have names or contact details of participants.

## Primary outcomes

Cognitive assessments from the original KEEPS-Cog [14] were replicated for the KEEPS Continuation Study. A battery of 11 cognitive tests were administered and summarized into 4 cognitive factor scores: verbal learning and memory (VLM), auditory attention and working memory (AAWM), visual attention and executive function (VAEF), and speeded language and mental flexibility (SLMF). Details on the derivation of factor scores have been previously published [14]. Additionally, global cognitive function was assessed with the Modified Mini-Mental State examination (3MSE) [20]. Data collection and analysis at the KEEPS Continuation visit focused on changes occurring during the interval between the participants' original KEEPS study participation and their KEEPS Continuation study involvement. Data available from the original KEEPS study included cognitive variables, demographic, biometric, and medical history data, and the carrier status for apolipoprotein E epsilon4 (*APOEε4*), a genetic risk factor for Alzheimer's disease [21].

## mHT use between end of KEEPS and recontact for KEEPS Continuation

Women were interviewed regarding their use of mHT after the end of KEEPS: "Have you taken hormone therapy since ending participation in the KEEPS trial?". Out of the 299 participants in KEEPS Continuation, 41 participants (overall, 13.71% of 299) continued with the mHT regimens used in the KEEPS trial or switched to another type of systemic mHT after the end of the study (oCEE, *n* = 17, 41.5% of 41; tE2, *n* = 15, 36.6% of 41; and placebo, *n* = 9, 22% of 41). Most of the KEEPS Continuation participants who went on to use systemic mHT or switched to another type of systemic mHT after the end of the study (*n* = 40 out of 275; oCEE, 42.5%; tE2, 35%; placebo, 22.5%) also had cognitive data in the original KEEPS trial.

## Statistical methods

All statistical analyses were performed with R software, Version 4.3.3 [22]. The analytical methods used to test the hypothesis in this study were determined prior to the start of the study. A detailed description of the planned analyses is provided as supporting information titled "**S1 Protocol**." To examine the influence of attrition, we compared baseline characteristics from the original KEEPS trial between women who participated and those who did not participate in the KEEPS Continuation, referred to here as nonparticipants (i.e., those to either

declined to participate or who were lost to follow-up). Characteristics were summarized using means and standard deviations for continuous variables and counts and percentages for categorical variables. Data from participants and nonparticipant characteristics were compared using Fisher exact test (for categorical variables) or Student *t* test (for continuous variables) as appropriate. For each statistical test, the type I error was set at 5% and the tests were 2 sided.

**Primary analyses: Latent growth model with a distal outcome.** The time interval between KEEPS randomization and KEEPS Continuation visits varied from 8 to 14 years across participants. We note that our original analysis plan included a linear mixed effect modeling approach. We used a latent growth model (LGM) analysis instead (see **S1 Methods** for more information). LGM with a distal (long-term) outcome were estimated to investigate whether participants' baseline cognition and changes in cognition across original KEEPS visits (growth factors) predicted cognitive performance 8 to 14 years later and whether mHT randomization modified this relationship [23–25]. Changes across time were modeled as linear, reflecting unequally spaced time points fixed at 0, 18, 36, and 48. The latent intercept and slope factors were allowed to correlate. The influence of mHT assignment was incorporated via direct effects on both the intercept and slope factors of cognitive performance during KEEPS trial and the distal outcome. LGM models were fitted separately for all the 4 cognitive factor outcomes and the global cognitive outcome measured by the 3MSE. All LGM models controlled for education, age, and *APOEe4* carrier status.

Model fit was evaluated using multiple indices: the (standardized) root mean square residual (SRMR) and comparative fit index (CFI) [26–28]. Values of the CFI $\geq 0.95$ and the SRMR $\leq 0.08$ were deemed to reflect good model fit [29]. To allow for estimations based on all available data and produce more efficient and less biased parameter estimates in the presence of non-normality and missing data, all LGM model parameters were estimated using robust maximum likelihood estimation procedures [30,31].

**Post hoc sensitivity analyses.** To clarify effects of mHT withdrawal, we conducted a post hoc sensitivity analysis excluding the 40 participants who had either continued the mHT regimens used in the KEEPS trial or switched to another type of systemic mHT after the end of the KEEPS trial. This exclusion was specifically directed to the primary goal of describing whether randomization to 4 years of mHT (in KEEPS) modified cognition 8 to 14 years later. For this reason, a simplified category of "any use" of systemically active mHT during the interval was used as an exclusionary criterion for these post hoc sensitivity analyses. Analyses were conducted separately for all 4 cognitive factor score outcomes and global cognition (3MSE).

**Secondary analyses: Cross-sectional comparison of cognitive outcomes.** Data from 292 participants were available to derive factors scores for women returning for a KEEPS Continuation visit. Using these data, we assessed differences in mean performance by KEEPS randomization groups across the 4 cognitive latent factor scores and global cognition using one-way analysis of variance models.

## Results

### Participant characteristics

**Table 1** provides a summary of participant characteristics at the time of enrollment into the KEEPS for the 275 women enrolled in the KEEPS Continuation. Also included in **Table 1** is a comparison of baseline characteristics between women who returned for the KEEPS Continuation and the non-participants enrolled in the original KEEPS for whom we had KEEPS baseline cognitive data. In general, KEEPS baseline characteristics of women returning for the KEEPS Continuation were similar to those not returning with the exception of blood pressure readings and baseline 3MSE scores.

**Table 1. Baseline characteristics from the original KEEPS trial in participants and nonparticipants of the KEEPS Continuation.** Characteristics represent original KEEPS trial data collected between years 2005 and 2008.

| Variable | KEEPS Continuation Nonparticipants (*n* = 387) KEEPS Baseline characteristics | KEEPS Continuation Participants with Cognitive Data (*n* = 275) KEEPS Baseline characteristics | *p*-Value |
|---|---|---|---|
| Age[a] at entry into KEEPS trial (mean, SD) | 52.556 (2.706) | 52.789 (2.427) | 0.254 |
| Time since Menopause in years (mean, SD) | 1.410 (0.706) | 1.458 (0.760) | 0.203 |
| Waist/Hip ratio (mean, SD) | 0.820 (0.079) | 0.812 (0.085) | 0.206 |
| Systolic BP (mm Hg) (mean, SD) | 119.952 (15.367) | 117.219 (14.303) | **0.021** |
| Diastolic BP (mm Hg) (mean, SD) | 75.444 (9.213) | 74.018 (8.81) | **0.047** |
| Glucose (mg/dL) (mean, SD) | 89.111 (10.179) | 89.178 (9.168) | 0.931 |
| Insulin (mcU/mL) (mean, SD) | 6.099 (8.211) | 6.203 (9.736) | 0.883 |
| HOMA-IR[b] (mean, SD) | 1.301 (2.553) | 1.26 (2.127) | 0.830 |
| 3MSE (mean score out of 100 total possible points, SD) | 96.37 (4.607) | 96.96 (3.716) | **0.044** |
| Education level, *n* (%) | | | |
| Some High School | 2 (0.522 %) | 1 (0.365 %) | 0.850 |
| High School Diploma or GED | 25 (6.527 %) | 21 (7.664 %) | |
| Some College/Vocational School | 78 (20.366 %) | 45 (16.423 %) | |
| College Graduate | 151 (39.426 %) | 115 (41.971 %) | |
| Some Graduate or Professional School | 17 (4.439 %) | 13 (4.745 %) | |
| Graduate or Professional Degree | 110 (28.721 %) | 79 (28.832 %) | |
| *APOE ε4* allele carrier, *n* (%) | 84 (24.46%) | 66 (26.40%) | 0.797 |
| Current or past smoker, *n* (%) | 100 (25.907%) | 56 (20.364%) | 0.098 |

Characteristics represent original KEEPS trial data collected between years 2005 and 2008 before the time when women were recontacted for the KEEPS continuation. Data shown are mean (SD) or *n* (%). *P*-values are from Fisher exact test or Student *t* test as appropriate. The statistically significant findings are in bolded font. BP, blood pressure; HOMA-IR, homeostasis model assessment of insulin resistance; 3MSE, Mini-Mental State Examination; KEEPS, Kronos Early Estrogen Prevention Study.
[a]Age was recorded and shared by participant's report of their age and not date of birth in KEEPS. Age was calculated based on date of birth in KEEPS Continuation.
[b]Log-transformed HOMA-IR.

Women who did not return for re-evaluation had a slightly but significantly higher baseline systolic blood pressure (*p* = 0.021). Differences in diastolic blood pressure and 3MSE scores were marginally significant (*p* = 0.047 and *p* = 0.044, respectively).

## Linear latent growth models

**Table 2** presents the model fit indices and parameter estimates for analyses conducted for each of the 5 primary outcomes. An examination of the fit indices suggests that the models provide a good fit to the data and that mHT randomization did not differentially influence cognitive outcomes during mHT nor when measured approximately 10 years after the end of the KEEPS trial. Specifically, women randomized to oCEE or tE2 demonstrated cognitive trajectories similar to those of women randomized to placebo in KEEPS. This was the case for all 4 cognitive factors and for performance on the global cognitive measure. Confirming findings from previous study analyses, during 4 years of randomization, the slope or growth curve for cognitive function on the 4 cognitive factors and the 3MSE (global cognition) was similar for women randomized to either form of mHT, oCEE, or tE2, when compared to those randomized to placebo. Follow-up at KEEPS Continuation suggested a similar null effect of randomization. Rather than mHT randomization, cognitive performance at the KEEPS Continuation visit (distal outcome) appeared to be more closely associated with performance at KEEPS baseline and cognitive performance over KEEPS study visits. That is, the strongest predictor of

**Table 2. Linear latent growth models for cognitive outcomes showing the association between intercept and slope for cognitive performance during menopausal hormone therapy and later cognitive function measured at KEEPS Continuation.**

| Variable | Estimate | S.E. | *P*-value | 95% Confidence Intervals |
|---|---|---|---|---|
| **Verbal Attention and Executive Function** | | | | |
| Intercept for cognitive performance | 0.396 | 0.055 | **<0.001** | (0.289–0.504) |
| Slope for cognitive performance | 0.528 | 0.056 | **<0.001** | (0.418–0.638) |
| *Effect of mHT allocation during KEEPS trial on slope for cognitive performance* | | | | |
| oCEE | 0.011 | 0.050 | 0.832 | (−0.088–0.110) |
| tE2 | −0.022 | 0.050 | 0.657 | (−0.121–0.077) |
| *Effect of mHT allocation during KEEPS trial on later cognitive function* | | | | |
| oCEE | −0.012 | 0.04 | 0.760 | (−0.090–0.066) |
| tE2 | −0.046 | 0.041 | 0.254 | (−0.126–0.033) |
| **Fit Indices** | | | | |
| CFI = 0.970; SRMR = 0.058 | | | | |
| AIC = 6,813.782; BIC = 6,931.443 | | | | |
| **Speeded Language and Mental Flexibility** | | | | |
| Intercept for cognitive performance | 0.480 | 0.075 | **<0.001** | (0.334–0.627) |
| Slope for cognitive performance | 0.476 | 0.096 | **<0.001** | (0.289–0.664) |
| *Effect of mHT allocation during KEEPS trial on slope for cognitive performance* | | | | |
| oCEE | −0.005 | 0.052 | 0.924 | (−0.108–0.098) |
| tE2 | −0.013 | 0.052 | 0.800 | (−0.114–0.088) |
| *Effect of HT allocation during KEEPS trial on later cognitive function* | | | | |
| oCEE | −0.021 | 0.037 | 0.579 | (−0.094–0.052) |
| tE2 | −0.032 | 0.041 | 0.437 | (−0.112–0.048) |
| **Fit Indices** | | | | |
| CFI = 0.955; SRMR = 0.064 | | | | |
| AIC = 7,501.294; BIC = 7,534.425 | | | | |
| **Auditory Attention and Working Memory** | | | | |
| Intercept for cognitive performance | 0.593 | 0.045 | **<0.001** | (0.505–0.681) |
| Slope for cognitive performance | 0.400 | 0.046 | **<0.001** | (0.311–0.489) |
| *Effect of mHT allocation during KEEPS trial on slope for cognitive performance* | | | | |
| oCEE | −0.062 | 0.057 | 0.279 | (−0.173–0.050) |
| tE2 | −0.042 | 0.056 | 0.445 | (−0.152–0.067) |
| *Effect of mHT allocation during KEEPS trial on later cognitive function* | | | | |
| oCEE | 0.011 | 0.047 | 0.817 | (−0.082–0.104) |
| tE2 | 0.052 | 0.042 | 0.222 | (−0.031–0.134) |
| **Fit Indices** | | | | |
| CFI = 0.960; SRMR = 0.047 | | | | |
| AIC = 6,053.908; BIC = 6,171.570 | | | | |
| Variable | Estimate | S.E. | P-value | 95% Confidence Intervals |
| **Verbal Learning & Memory** | | | | |
| Intercept for cognitive performance | 0.418 | 0.051 | **<0.001** | (0.319–0.517) |
| Slope for cognitive performance | 0.455 | 0.063 | **<0.001** | (0.331–0.579) |
| *Effect of mHT allocation during KEEPS trial on slope for cognitive performance* | | | | |
| oCEE | −0.076 | 0.055 | 0.167 | (−0.183–0.032) |
| tE2 | −0.068 | 0.053 | 0.203 | (−0.172–0.036) |
| *Effect of mHT allocation during KEEPS trial on later cognitive function* | | | | |
| oCEE | −0.084 | 0.049 | 0.086 | (−0.181–0.012) |
| tE2 | −0.076 | 0.051 | 0.138 | (−0.177–0.024) |

(*Continued*)

**Table 2.** (Continued)

| Variable | Estimate | S.E. | P-value | 95% Confidence Intervals |
|---|---|---|---|---|
| **Fit Indices** | | | | |
| CFI = 0.980; SRMR = 0.038 | | | | |
| AIC = 8,874.730; BIC = 8,992.392 | | | | |
| **Global Cognition (Modified Mini-Mental State Test)** | | | | |
| Intercept for cognitive performance | 0.951 | 0.219 | **<0.001** | (0.522–1.380) |
| Slope for cognitive performance | 0.421 | 0.334 | 0.207 | (−0.233–1.076) |
| *Effect of mHT allocation during KEEPS trial on slope for cognitive performance* | | | | |
| oCEE | 0.170 | 0.137 | 0.214 | (−0.098–0.439) |
| tE2 | 0.100 | 0.107 | 0.350 | (−0.110–0.310) |
| *Effect of mHT allocation during KEEPS trial on later cognitive function* | | | | |
| oCEE | 0.0001 | 0.06 | 0.995 | (−0.118–0.017) |
| tE2 | −0.103 | 0.064 | 0.106 | (−0.228–0.022) |
| **Fit Indices** | | | | |
| CFI = 0.991; SRMR = 0.031 | | | | |
| AIC = 12,684.378; BIC = 12,742.698 | | | | |

The statistically significant findings are in bold font. mHT, menopausal hormone therapy; KEEPS, Kronos Early Estrogen Prevention Study; oCEE, oral conjugated equine estrogens; tE2, transdermal estradiol; S.E, standard error; CFI, Comparative Fit Index; AIC, Akaike Information Criterion; BIC, Bayesian Information Criterion; SRMR, Standardized Root Mean Square Residual.

cognitive performance in KEEPS Continuation was cognitive performance in KEEPS trial both at baseline (intercept) and across time (slope).

## Post hoc sensitivity analyses

Of those (had cognitive data available both from KEEPS and KEEPS Continuation) who indicated any use of mHT after the KEEPS ended most had been randomized to an active arm during the KEEPS; oCEE, *n* = 17 (42.5%); tE2, *n* = 14 (35%); placebo, *n* = 9 (22.5%). The results of the sensitivity analyses after removing participants (*n* = 40) who continued the use of systemic mHT or switched to another type of systemic mHT or started to use a systemic mHT (from placebo group) after the end of KEEPS are presented in the **S1 Table**. The overall impact on parameter estimates was relatively trivial.

## Cross-sectional comparison of cognitive outcomes

As shown in **Table 3**, the comparison of cognitive performance at the KEEPS Continuation visit revealed no significant group differences between the mHT and placebo groups on the 4 cognitive factors and global cognition. *P*-values for the factor scores were all >0.40 and effect sizes ($\eta^2$) < 0.006. Although the *P*-value for the 3MSE was marginal (*p* = 0.059), differences did not reach statistical significance for this global cognitive measure. **Fig 3** illustrates the consistent trends in cognitive performance by HT group across all factor scores and global cognition. Additionally, a comparison between the proportion of participants (*n* = 257) using systemic mHT after KEEPS did not reach statistical significance among the treatment groups (*p* = 0.097).

## Discussion

The primary aim of the KEEPS Continuation Study was to examine whether prior exposure to mHT was associated with long-term and lasting effects on cognition. This was not addressed

**Table 3. Cross-sectional comparisons of cognitive performance at the time of KEEPS Continuation.**

| Cognitive Performance | Sum of Squares | df | MS | F | p | $\eta^2$ |
|---|---|---|---|---|---|---|
| *Verbal Learning and Memory* | | | | | | |
| Treatment Group | 0.661 | 2 | 0.331 | 0.36 | 0.698 | 0.005 |
| Residuals | 265.109 | 289 | 0.917 | | | |
| *Speeded Language and Mental Flexibility* | | | | | | |
| Treatment Group | 0.304 | 2 | 0.152 | 0.18 | 0.835 | 0.002 |
| Residuals | 243.511 | 289 | 0.843 | | | |
| *Auditory Attention and Working Memory* | | | | | | |
| Treatment Group | 0.593 | 2 | 0.296 | 0.38 | 0.684 | 0.003 |
| Residuals | 225.432 | 289 | 0.78 | | | |
| *Visual Attention and Executive Function* | | | | | | |
| Treatment Group | 1.486 | 2 | 0.743 | 0.877 | 0.417 | 0.004 |
| Residuals | 244.889 | 289 | 0.847 | | | |
| *Global Cognition (3MS)* | | | | | | |
| Treatment Group | 62.238 | 2 | 31.119 | 2.857 | 0.059 | 0.020 |
| Residuals | 3,202.315 | 294 | 10.892 | | | |

df, Degrees of Freedom; MS, Mean Square; F, F-statistic; p, p-value; η^2, Eta Squared.

earlier in the original KEEPS or its ancillary KEEPS-Cog study, which examined the short-term effects of mHT. In the present study, approximately 10 years after 48 months of early menopausal therapy, i.e., mHT in KEEPS trial, the cognitive performance of women randomized to oCEE or tE2 did not differ from those randomized to placebo. This was demonstrated as an outcome relative to KEEPS baseline performance and cognitive performance over time while enrolled in the KEEPS, and when cognition at the KEEPS Continuation was compared cross-sectionally across the KEEPS randomization groups. Thus, contrary to our hypothesis, there appears to be no long-term beneficial or harmful cognitive effects of HT use when initiated around the time of menopause.

These data counter concerns for cognitive harms associated with mHT from the WHIMS study [9–11]. It is important to point out that women enrolled in WHIMS were all age 65 years or older at the time of enrollment and HT randomization, with a mean age of 69 at baseline compared to women in KEEPS whose mean age was 52 years when randomized to mHTs or placebo. In addition, some women in WHIMS had increased CVD risk at baseline, whereas women in KEEPS were at low risk for CVD. In other words, the WHIMS did not investigate effects of early postmenopausal HT on cognitive outcomes. Rather, the study offered important insights on the cognitive effects of late postmenopausal HT.

The KEEPS Continuation finding of no long-term influence on cognitive performance adds to our understanding of the safety of mHT use in early postmenopausal women who are at low CVD risk. Moreover, these data contribute to the emerging understanding of the hypothetical "critical window" for mHT use. Briefly, after the surprising findings from the WHIMS were published, it was theorized that the peri- or early postmenopause period was a critical window [32] for HT use, and that mHT could result in both deleterious or beneficial effects on cognitive health, depending on the timing of administration [33]. Biological explanations for timing theories linked time-since-menopause to the health of underlying cells and substrates (i.e., the healthy cell bias and intact mitochondrial bioenergetics) [34].

Several studies partly tested the critical window hypothesis for brain health by examining the short-term effects of mHT, including the KEEPS-Cog Study [14], an ancillary study to the

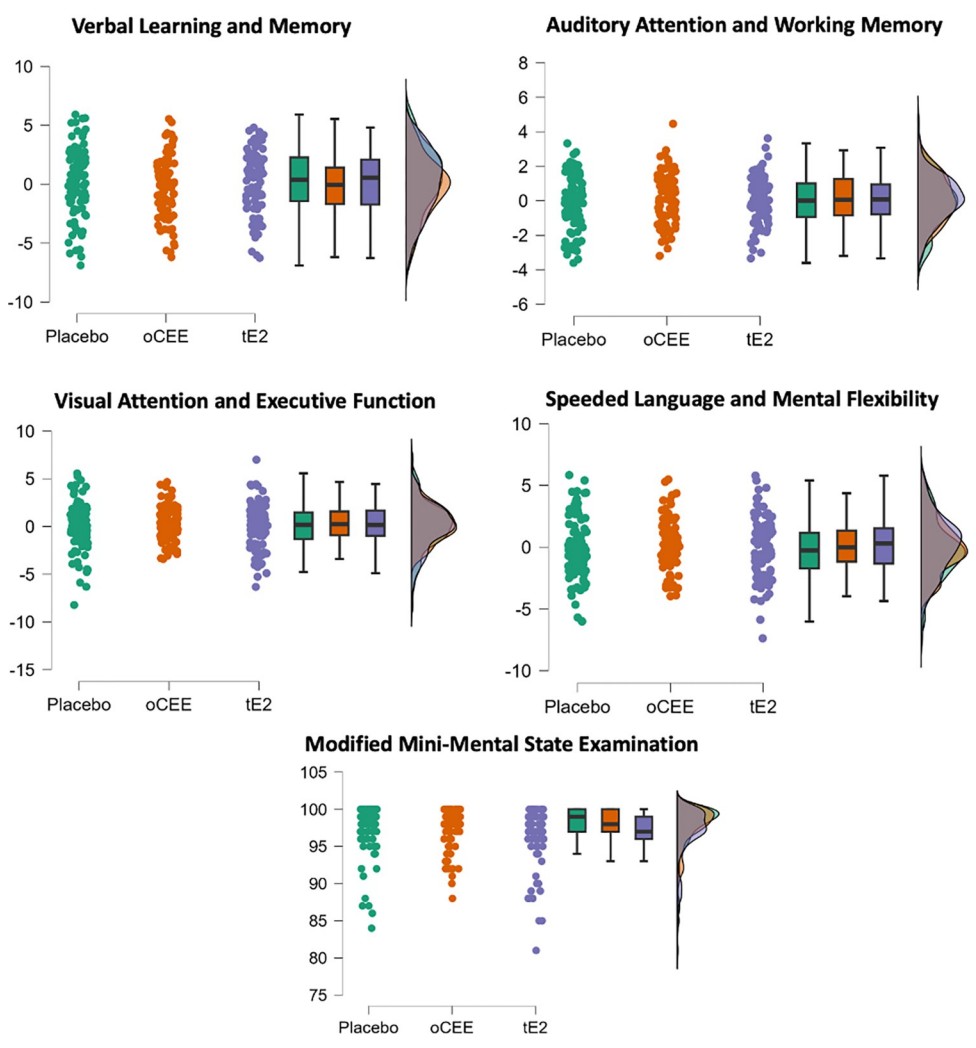

**Fig 3. Cross-sectional Cognitive results.** Distribution of cognitive factor scores and the modified mini-mental state examination (global cognition) by treatment group in the KEEPS Continuation study. oCEE, oral conjugated equine estrogens; tE2, transdermal estradiol. Total possible score was 100 points on the Modified Mini-Mental State Examination (3MS).

KEEPS. Findings from the KEEPS-Cog suggested that women treated with tE2 plus micronized progesterone, oCEE plus micronized progesterone, or placebo exhibited no significant cognitive benefits or harms on 4 cognitive domains and a global cognitive measure with 48 months of therapy. Notably, participants who received oCEE reported fewer depression and anxiety symptoms over the course of 4 years compared to those on placebo. Another randomized control trial, the Early versus Late Intervention Trial with Estradiol (ELITE) [15], administered an oral estradiol for up to 5 years to both women who were within 6 years of menopause and women who were 10 or more years past menopause. Like KEEPS-Cog, ELITE findings revealed no cognitive harm or benefit for younger women. In contrast to WHIMS, the ELITE data also suggested no harm or benefit on cognition for the women randomized to HT a decade after onset of menopause. The KEEPS Continuation findings extend our understanding of the cognitive effects of mHT beyond short-term effects; specifically, use of HT within the time frame of early menopause appears to have no long-term cognitive effects.

To explore the critical window hypothesis, the WHIMS investigators re-examined their data, also concluding that HT use proximal to menopause was not associated with long-term cognitive effects. The WHIMS of Younger Women (WHIMSY) [35] re-evaluated women enrolled in the WHI oCEE-alone or CEE+MPA trials when they were between the ages 50 and 55, approximately 7 years after their discontinuation of HT, or placebo [35,36]. However, cognitive data from these younger cohort were not collected during the participants' active intervention phases with study medications. Thus, cognitive performance could only be compared cross-sectionally. Participants receiving mHT at ages 50 to 55 showed no differences in cognitive performance compared to women randomized to placebo. KEEPS Continuation data provide consistent evidence that both mHT groups performed similarly to placebo on cognitive measures approximately 10 years after they were randomized to either mHT or placebo. Our KEEPS Continuation models add to our understanding by examining cognition performance indicators prior to and during mHT exposure. Overall, models showed strong associations between baseline and change-in-cognition during KEEPS and the same measures in KEEPS Continuation, i.e., strongest predictor of cognitive performance in KEEPS Continuation was cognitive performance in KEEPS.

Altogether, these findings add to our understanding of the importance of timing of mHT administration, but do not fully support the original critical window hypothesis for cognition. Likewise, our KEEPS Continuation hypothesis that tE2 would demonstrate cognitive benefits over placebo, and oCEE would show no difference from placebo was not supported. Specifically, while exposure to specific forms of mHT well past menopause appear to be associated with cognitive harms, mHT in early postmenopause demonstrates neither harm nor benefits.

KEEPS neuroimaging studies investigating structural and metabolic changes following mHT supported the importance of timing, but also highlighted continued discrepancies in the extant literature—possibly related to form of mHT. For example, in another single-site ancillary KEEPS study focused on neuroimaging, Kantarci and colleagues [37] demonstrated that tE2 but not oCEE was associated with decreased amyloid β deposition on PET, especially in *APOEε4* carriers. Furthermore, women treated with both forms of mHT had increased ventricular volume, when receiving oCEE and tE2 compared to women randomized to placebo, but the difference was significant only in the oCEE group [38]. However, this increase in ventricular volume was no longer present 7 years post-randomization (i.e., 3 years after the end of the 4-year mHT trial), suggesting that this may be a physiological effect only present during the mHT phase [16]. The authors noted that initiation of mHT may have to fall within 3 years of menopause in order to reduce the risk of ventricular expansion. Furthermore, there was a better preservation of the prefrontal cortex volume in tE2 group which was associated with lower amyloid deposition. In KEEPS Continuation, approximately 14 years post-randomization, we expected some long-term cognitive benefits for participants randomized to tE2. However, we found no advantage for either form of mHT over placebo. It is possible that initial structural changes may not translate directly to observable cognitive differences more than a decade later, particularly given the complex interplay of aging, genetics, and other environmental factors. Additionally, the dose and duration of mHT in KEEPS may have been insufficient to elicit marked cognitive effects, especially considering that cognitive decline in healthy postmenopausal women can be subtle or transient. As outlined by Morgan and colleagues and Greendale (2010, 2011), 2 key longitudinal cohort studies—Kinmen women-health investigation (KIWI) (Fuh and colleagues) and Study of Women Across the Nation (SWAN) (Greendale and colleagues)—have examined cognitive changes during menopause [39–43]. These observational studies described subtle cognitive declines occurring during the menopausal transition, typically manifesting as reduced learning effects over time rather than significant performance drops. Finally, the absences of a long-term effect of the early differential

structural changes associated with the 2 mHT formulations might reflect a ceiling effect, where the relatively healthy and well-educated study population performed near their cognitive potential, leaving little room for measurable improvements or declines. Altogether, early data pointing toward a differential effect of tE2 over placebo were not supported by the KEEPS Continuation findings. The KEEPS Continuation included neuroimaging data collection. Findings in this larger sample will clarify the long-term effects of mHT on Alzheimer's disease biomarkers and brain structure.

Importantly, multiple key study design features and participant characteristics may contribute to the findings in the KEEPS Continuation—all of which may reduce risk for cognitive declines in the KEEPS participants. Not only were all women in the KEEPS baseline within 3 years of their final menstrual period and with low CVD risk, but also none were diabetic, and none had undergone hysterectomy. The KEEPS utilized a lower oCEE dose than that used in the WHIMS, and unlike WHIMS included a tE2 formulation randomization group. Finally, estrogen was administered with cyclic micronized progesterone in an attempt to mimic a more physiologic paradigm, while the WHIMS trial utilized continuous administration of synthetic progestin (MPA). These key differences in study design and study population likely contributed in some part to the discrepant findings between the KEEPS-Cog and KEEPS Continuation, and the WHIMS.

KEEPS Continuation was not without limitations. Only 299 out of the 727 of the original KEEPS cohort (41%) participated in the KEEPS Continuation. A significant portion of this recruitment occurred during the COVID-19 pandemic (2020 to 2022), which severely hindered enrollment efforts. The pandemic led to reluctance among participants to travel or attend the study sites, and many procedures were delayed or canceled due to institutional closures, creating numerous scheduling challenges. Despite these obstacles, the KEEPS Continuation successfully recruited 299 participants, a substantial number for a follow-up study of a clinical trial that randomized participants to an intervention up to 14 years earlier. Of these 299 participants, the majority (275 or 92%) had KEEPS trial data on cognitive outcomes. Data regarding the type, dose, and duration of mHT used post KEEPS were self-reported, raising concerns about potential for recall bias. Although our analyses relied only on the simplified report of any use of systemic HT versus no use, the effects of the more granular and often imprecise reports about formulation, dose, and length of use were not assessed. Differences in baseline global cognition and SBP between participants and non-participants may reflect a healthy survivor bias, although some non-participation was due to factors not typically encountered in longitudinal follow-up studies, e.g., the COVID-19 pandemic, missing contact information because the PI moved to another institution or passed away. Participants of KEEPS Continuation were primarily non-Hispanic and white, generally well educated and by KEEPS design, free from many comorbid conditions; thus, population characteristics limit the generalizability of results to more racially and ethnically diverse populations who have varying level of education and health status. In particular, KEEPS baseline participants had low CVD risk. So, the results may not be generalizable for those with greater CVD risk. Finally, like other research involving a prolonged period of time between original study involvement and follow-up, the participants who returned for the KEEPS Continuation study were likely more advantaged than those who did not return for follow-up—often in unmeasured factors that influence cognition (income, geographic settings, employment, family support).

While KEEPS Continuation encountered challenges due to attrition, it is worth noting that the study's design may have mitigated the full impact of a healthy survivor bias. The original cohort was selected for its low cardiovascular disease risk, and exclusions for conditions like diabetes, elevated BMI, and tobacco use likely contributed to the healthier profile of participants at baseline. This may have reduced the bias typically expected in long-term studies.

Moreover, while the differences in baseline global cognition and systolic blood pressure (SBP) between participants and non-participants were aligned with a healthy survivor bias, the unique circumstances surrounding non-participation—such as the COVID-19 pandemic and the relocation or passing of site PIs—should also be acknowledged as contributing factors beyond typical health-related attrition. Additionally, investigating the full effects of attrition, such as stroke history or cause of death in non-responders, was not feasible due to study constraints, as participant identifiers were held by individual sites and the study teams had been disbanded. This limitation highlights the challenges in fully understanding the effects of attrition in long-term follow-up studies.

The WHIMS findings and WHI data showing increased risk for cardiovascular events and breast cancer [7] led to dramatic shifts in mHT use. Prevalence of use declined to nearly half of pre-WHI levels in the first years after findings were published [44], but marginally rebounded in subsequent years [45]. In the decades since the WHI publications, women using HT at menopause do so at lower doses and in a greater variety of forms and routes of administration [46]. In general, data from the Study of Women Across the Nation suggest that women entering menopause are avoiding HT altogether, even when they are symptomatic [45].

In contrast, mHT is an effective therapy for menopause associated symptoms, e.g., vasomotor symptoms [5]. Data presented here add to the accumulating evidence from several clinical trials, including KEEPS and KEEPS-Cog, which point toward mHT in early menopause being safe for both short- and long-term cognitive health. On the other hand, there are health risks associated with mHT use, including possibility of certain cancers. This can be mitigated by limiting the dosage, length of time, and possibly the route and formulation of mHT that is administered, although randomized trials of these approaches are lacking.

A tailored or precision medicine approach would be optimal, allowing clinicians to fully understand characteristics that would make it unsafe for women to initiate therapy. Espeland and colleagues [47] found that among women enrolled in the WHIMS, those with diabetes who were randomized to oCEE demonstrated the highest risk of developing cognitive impairment and probable dementia compared to those without diabetes regardless of mHT assignment. Likewise, data from the KEEPS highlighted the need to further characterize pharmacogenomic interactions, describing how genetic variants appeared to interact with mHT status to affect cardiovascular phenotypes [48].

Altogether, data are still needed to guide the healthcare of women entering the menopausal transition. Specifically, data to assist women in making personalized, informed decisions regarding management of their menopausal symptoms and the prevention of future adverse health outcomes. Ideally, a woman seeking to manage symptoms occurring during early postmenopause with HT would have specific and personalized guidance, such that she need not carry undue concerns, or be unaware of real risks should she opt to use HT to manage her menopausal symptoms.

## Conclusions

Approximately a decade after randomization, women treated with 4 years of mHT performed similarly on cognitive factors assessing 4 domains and a global cognitive measure to women treated with placebo. Findings may reassure women opting to use hormone therapy in early menopause, to manage menopausal symptoms, that 4 years of therapy started within 3 years of menopause had no long-term deleterious impact on cognition. Our results also indicate that mHT does not prevent cognitive decline when initiated around the time of menopause. Therefore, mHT should not be recommended as a strategy for improving or preserving cognitive function in recently menopausal women with low cardiovascular risk.

## Supporting information

**S1 STROBE Checklist. STROBE Statement.**
(DOCX)

**S1 Protocol. Prospective Observational Study.**
(DOCX)

**S1 Methods. Statistical Methods.**
(DOCX)

**S1 Table. Linear latent growth models for cognitive outcomes showing the association between intercept and slope for cognitive performance during hormone therapy and later cognitive function after excluding n = 40 participants who continue the use of systemic mHT.**
(DOCX)

## Acknowledgments

The authors gratefully acknowledge the study participants and staff of the KEEPS Continuation for their time and effort.

## Author Contributions

**Conceptualization:** Carey E. Gleason, N. Maritza Dowling, Sherman M. Harman, JoAnn E. Manson, Dustin B. Hammers, Frederick N. Naftolin, Lubna Pal, Virginia M. Miller, Marcelle I. Cedars, Rogerio A. Lobo, Michael Malek-Ahmadi, Kejal Kantarci.

**Data curation:** Carey E. Gleason, Taryn T. James, Hector Salazar, Carola A. Ferrer Simo, Sherman M. Harman, JoAnn E. Manson, Dustin B. Hammers, Frederick N. Naftolin, Lubna Pal, Virginia M. Miller, Marcelle I. Cedars, Rogerio A. Lobo, Michael Malek-Ahmadi, Kejal Kantarci.

**Formal analysis:** Carey E. Gleason, N. Maritza Dowling, Firat Kara, Kejal Kantarci.

**Funding acquisition:** Carey E. Gleason, N. Maritza Dowling, Firat Kara, Taryn T. James, Sherman M. Harman, JoAnn E. Manson, Dustin B. Hammers, Frederick N. Naftolin, Lubna Pal, Virginia M. Miller, Marcelle I. Cedars, Rogerio A. Lobo, Michael Malek-Ahmadi, Kejal Kantarci.

**Investigation:** Carey E. Gleason, N. Maritza Dowling, Firat Kara, Taryn T. James, Sherman M. Harman, Kejal Kantarci.

**Methodology:** Carey E. Gleason, Hector Salazar, Carola A. Ferrer Simo, Sherman M. Harman, Kejal Kantarci.

**Project administration:** Carey E. Gleason, Taryn T. James, Hector Salazar, Carola A. Ferrer Simo, Kejal Kantarci.

**Resources:** Carey E. Gleason, Sherman M. Harman, JoAnn E. Manson, Dustin B. Hammers, Frederick N. Naftolin, Lubna Pal, Virginia M. Miller, Marcelle I. Cedars, Rogerio A. Lobo, Michael Malek-Ahmadi, Kejal Kantarci.

**Supervision:** Carey E. Gleason, Taryn T. James, Hector Salazar, Carola A. Ferrer Simo, Kejal Kantarci.

**Writing – original draft:** Carey E. Gleason, N. Maritza Dowling, Firat Kara.

**Writing – review & editing:** Carey E. Gleason, N. Maritza Dowling, Firat Kara, Taryn T. James, Hector Salazar, Carola A. Ferrer Simo, Sherman M. Harman, JoAnn E. Manson, Dustin B. Hammers, Frederick N. Naftolin, Lubna Pal, Virginia M. Miller, Marcelle I. Cedars, Rogerio A. Lobo, Michael Malek-Ahmadi, Kejal Kantarci.

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
