## [Editor Report · Decision Letter 0]

27 Jun 2024

Dear Dr Gleason, 

Thank you for submitting your manuscript entitled "Long-term cognitive effects of menopausal hormone therapy: Findings from the KEEPS Continuation Study" for consideration by PLOS Medicine.

Your manuscript has now been evaluated by the PLOS Medicine editorial staff and I am writing to let you know that we would like to send your submission out for external peer review.

Please re-submit your manuscript within two working days, i.e. by Jul 01 2024 11:59PM. However, please do let us know if you need more time.

Feel free to email the editorial office at plosmedicine@plos.org if you have any queries relating to your submission and uploading the metadata, etc. For anything else, please do not hesitate to contact me directly (ssunny@plos.org).

Kind regards,

Syba Sunny, MBBS, MRes, FRCPath

Associate Editor

PLOS Medicine

ssunny@plos.org

---

## [Decision Letter · Decision Letter 1]

13 Aug 2024

Dear Dr. Gleason,

Many thanks for submitting your manuscript " Long-term cognitive effects of menopausal hormone therapy: Findings from the KEEPS Continuation Study" (PMEDICINE-D-24-02037R1) for consideration at PLOS Medicine. 

Your paper has now been reviewed by subject reviewers and a statistician; the comments are included below. It was also discussed with an academic editor with relevant expertise and the wider editorial team. 

As you will see, the reviewers were largely very positive about your manuscript, but offered some suggestions that could strengthen the paper. In addition, an academic editor with relevant expertise read through your paper and added some comments to this. After discussing the paper with the editorial team, I'm pleased to invite you to revise the paper in response to the comments made by the reviewers and the academic editor. Please address all of these in full. Please also note that we plan to send the revised paper to some or all of the original reviewers, and we cannot provide any guarantees at this stage regarding publication. 

When you upload your revision, please include a point-by-point response that addresses all of the reviewer and editorial points, indicating the changes made in the manuscript and either an excerpt of the revised text or the location (e.g. page and line number) where each change can be found. Please submit a clean version of the paper as the main article file and a version with changes marked should as a marked-up manuscript. Please also check the guidelines for revised papers at http://journals.plos.org/plosmedicine/s/revising-your-manuscript for any that apply to your paper.

We ask that you submit your revision by Sep 03 2024 11:59PM. However, if this deadline is not feasible, please contact me by email (ssunny@plos.org), and we can discuss a suitable alternative.

Please don’t hesitate to contact me directly with any questions.

Kind regards,

Syba

Syba Sunny MBBS, MRes, FRCPath

Associate Editor 

PLOS Medicine

ssunny@plos.org

*Please note: If your article is accepted, you may have the opportunity to make the peer review history publicly available. The record will include editor decision letters (with reviews) and your responses to reviewer comments. If eligible, we will contact you to opt in or out.

Comments from the academic editor:

The academic editor read your manuscript with great interest. She agreed with the reviewer comments but added some of her own thoughts. Firstly, she felt that more needed to be done to address the issue that those who did not take part and who died had unknown cognitive decline. She wondered if the authors could do anything more to address this? She suggested that perhaps sensitivity analyses may be a useful approach, drawing on existing data from other studies where dropout has been shown to be associated with cognitive decline. She postulated that potentially it may be that, at one extreme end, the reasons for dropout / decline might actually reinforce the findings in this paper. But she was concerned that, in other possible scenarios, the dropout could possibly have skewed the results here. She also wondered whether the authors might be able to use death certification and recorded causes of death to help enlighten us on this issue.

Comments from the reviewers: 

Reviewer #1: This is such an important paper in this field and its results were long awaited. 

These renowned researchers also report that MHT close to age at menopause in healthy women does not confer risk of cognitive decline, also not after 10 year of follow-up (see also Vinogradova's paper and that of other National Registries). 

I may have missed this some of this (as very jet-lagged, my apologies) but am wondering if the authors had data on VMS/sleep/mood and whether that modified these effects or whether that is part of another paper? 

It seems that the n=275 left for FU after 10 years were not different from the baseline cohort apart from baseline global cognition and SBP and that should perhaps be a little more highlighted in the paper, as this is very important. Could that have affected outcomes? Would those with poor cognition and drop-out be more likely not to want to/be able to return? This group had the highest opportunity to improve over time so that could be healthy survivor bias perhaps? Also could stroke after MHT (with the higher BP) have resulted in those at risk dropping out not showing a potential worsening over time? 

The 41% of participants who continued on MHT, how where they different? Could a GLM be done using none-past and ongoing user category?

Lastly I often find that using compound scores somehow reduces sensitivity of tests to change. Could the recall score be analysed seperately rather than within the compound score?

Maybe all this is better suited for another paper?

Such an interesting read and well embedded in the literature. Very reassuring for white healthy middle-class educated women. As the authors said: more research needs to be done with low SEP and minority ethnicity groups 

Reviewer #2: Statistical review

This paper reports a long-term follow-up of a randomised trial of hormone therapy for pre-menopausal women. The aim of this paper is to assess whether use of therapy is associated with any long term cognitive function differences. The results indicate that there is no significant difference between groups for any of the cognitive outcomes. Although the proportion of original trial participants who took part in the longer-term study was not high, the sub-study population seems to be fairly well representative of the original trial population.

I found the paper to be well described and reported and only had some minor comments on the statistical methods/reporting:

1. Abstract line 49-50: can it be clarified whether there was 2 or 3 randomised groups (it wasn't clear to me whether the two mHTs were distinct treatment arms or there was one mHT arm which could then choose between mHTs.

2. Abstract: it would be good to provide estimated associations and between-group differences (together with 95% CIs and p-values) in the results section.

3. Was this study registered? If so please provide the registration information.

4. Page 14, line 262: I did not follow the percentages other than 'overall'. E.g. why is 41.5% given for oGEE if 41 is the total number across all arms?

5. Statistical methods: as latent growth models would probably not be well known to readers, it might be helpful to have supplementary material that describes the model and also what assumptions are made. 

6. Results: I'm not sure if the LGMs used naturally have any visualisation of results, but I would have found it interesting to see the mean trajectory over time in the different groups.

James Wason

Reviewer #3: Very important area of research. However, not novel 

1. The abstract can be made more concise and clearer with only necessary information especially the Methods and findings section..

2. The reason for the hypothesis that the two treatment formulations would affect cognition differentially, can be explained stronger. The absence of such a difference could also be explained with possible reasons in the discussion with appropriate references.

3. The criteria used to classify women as low CVD risk and the rationale for excluding those with high CVD risk could be elaborated.

4. What can be the possible reasons for the difference in impact of mHT and delayed HT on cognition?

5. Were all the participants cognitively healthy (absence of MCI or dementia? If not could the frequency of cognitive impairment also be compared among the groups?

6. Could the baseline difference in 3MSE scores have influenced the results?

7. The significance of performing the cross-sectional analysis on cognitive functioning can be explained better. Would a longitudinal study have made differences compared to the current methodology. 

8. A separate analysis for the people who continued HT (depending on duration of HT) could have added clarity to the time-since-menopause theories.

---

* Please upload any figures associated with your paper as individual TIF or EPS files with 300dpi resolution at resubmission; please read our figure guidelines for more information on our requirements: http://journals.plos.org/plosmedicine/s/figures. While revising your submission, please upload your figure files to the PACE digital diagnostic tool, https://pacev2.apexcovantage.com/. PACE helps ensure that figures meet PLOS requirements. To use PACE, you must first register as a user. Then, login and navigate to the UPLOAD tab, where you will find detailed instructions on how to use the tool. If you encounter any issues or have any questions when using PACE, please email us at PLOSMedicine@plos.org.

* Thank you for including a data availability statement. Could we ask if you could provide different contact details please? When possible, we recommend authors deposit restricted data to a repository that allows for controlled data access. If this is not possible, directing data requests to a non-author institutional point of contact, such as a data access or ethics committee, helps guarantee long term stability and availability of data. Providing interested researchers with a durable point of contact ensures data will be accessible even if an author changes email addresses, institutions, or becomes unavailable to answer requests. Further information is available here: https://journals.plos.org/plosmedicine/s/data-availability#loc-faqs-for-data-policy

FORMATTING & STYLE

* Thank you for providing an Author Summary of your research. 

The section ‘What did the researchers do and find?’ contains a good amount of information but reads a little too technical currently, and, thus, may not be well-understood by non-experts in this topic. The Author Summary should help to make your findings more accessible to as wide an audience as possible. So, could you please tweak these sentences to use less technical language? Whilst accuracy is important, the Author Summary can contain more simplified information.

Also, please revise your ‘What do these findings mean?’ section to comply with the following PLOS guidance:

‘Authors should reflect on the new knowledge generated by the research and the implications for practice, research, policy, or public health. Authors should also consider how the interpretation of the study’s findings may be affected by the study limitations. In the final bullet point of ‘What Do These Findings Mean?’, please describe the main limitations of the study in non-technical language.’

Please see our author guidelines for more information: https://journals.plos.org/plosmedicine/s/revising-your-manuscript#loc-author-summary

* Please continue to include page numbers and line numbers in the manuscript file. Use continuous line numbers (do not restart the numbering on each page). 

FIGURES AND TABLES

SUPPLEMENTARY MATERIAL

* Please ensure that all references to Supporting Information are made as outlined here: https://journals.plos.org/plosmedicine/s/supporting-information

OBSERVATIONAL STUDIES

* Abstract: Please include the study design, population and setting, number of participants, years during which the study took place (enrollment and follow up), length of follow up, and main outcome measures.

* Please ensure that the study is reported according to the STROBE (or appropriate STROBE extension) guideline (available from: https://www.equator-network.org/reporting-guidelines/strobe) and include the completed STROBE (or STROBE extension) checklist as Supporting Information. Please add the following statement, or similar, to the Methods: "This study is reported as per the Strengthening the Reporting of Observational Studies in Epidemiology (STROBE) guideline (S1 Checklist)." When completing the checklist, please use section and paragraph numbers, rather than page numbers. 

* For all observational studies, in the manuscript text, please indicate: (1) the specific hypotheses you intended to test, (2) the analytical methods by which you planned to test them, (3) the analyses you actually performed, and (4) when reported analyses differ from those that were planned, transparent explanations for differences that affect the reliability of the study's results. If a reported analysis was performed based on an interesting but unanticipated pattern in the data, please be clear that the analysis was data driven. 

* Please state in the Methods section whether the study had a prospective protocol or analysis plan. If a prospective analysis plan (from your funding proposal, IRB or other ethics committee submission, study protocol, or other planning document written before analyzing the data) was used in designing the study, please include the relevant document(s) with your revised manuscript as a Supporting Information file to be published alongside your study and cite it in the Methods section. A legend for this file should be included at the end of your manuscript. If no such document exists, please make sure that the Methods section transparently describes when analyses were planned, and when/why any data-driven changes to analyses took place. Changes in the analysis, including those made in response to peer review comments, should be identified as such in the Methods section of the paper, with rationale.

---

## [Decision Letter · Decision Letter 2]

8 Oct 2024

Dear Dr. Gleason,

Thank you very much for re-submitting your manuscript "Long-term cognitive effects of menopausal hormone therapy: Findings from the KEEPS Continuation Study" (PMEDICINE-D-24-02037R2) for review by PLOS Medicine.

I have discussed the paper with my colleagues and the academic editor and it was also seen again by 3 reviewers. I am pleased to say that provided the remaining editorial and production issues are dealt with we are planning to accept the paper for publication in the journal.

[LINK]

We expect to receive your revised manuscript within 2 weeks (extended given the number of requests that the authors need to address). However, please do let us know if you need more time.

We look forward to receiving the revised manuscript by Oct 22 2024 11:59PM. 

Sincerely,

Syba

Syba Sunny, MBBS, MRes, FRCPath

Associate Editor 

PLOS Medicine

ssunny@plos.org

COMMENTS FROM THE ACADEMIC EDITOR:

The academic editor looked through your rebuttal letter and thought that there was a lot of valuable information that could be incorporated into the main text of the manuscript for the benefit of the readers. For example, she suggests paraphrasing what you’d written in response to the request for information about attrition analyses to the paper’s existing discussion of the study’s limitations, i.e. to explain why this wasn’t feasible, etc. 

She also said the following: ‘They need to have a summary that reflects not only that harm isn't done but that cognition is not improved as there is still medication being recommended in some places to prevent cognitive decline - the findings suggest no effect so this needs to be considered in both directions of earlier proposed impact.’ Our suggestion is that the authors revise the last couple of lines of the Abstract to reflect this, as well as add this point to the Discussion and/or Conclusion. 

COMMENTS FROM REVIEWERS:

Reviewer #1: I think this important paper has been improved by revision and limitations are well addressed and described. These data add to a very important discussion for clinicians and menopausal women on whether short term use (<5 years) confers later life accelerated cognitive decline. The data show that is unlikely to be the case given the limitations of possible healthy survivor bias

Re competing interest, I reviewed similar data for NICE on national guidelines for HRT and dementia risk (unpaid as expert) but do not work with these authors.

Reviewer #2: Thank you to the authors for addressing my previous comments well. I note the issues with adding in the requested results to the abstract.

Reviewer #3: Dear Author,

Thank you for incorporating the changes suggested.

regards

COMMENTS FROM THE EDITORS:

Thank you for submitting a revised manuscript. We appreciate the significance of your work and are pleased that your submission is moving closer towards publication. Please note that it appears that several of our previous editorial requests were overlooked in your revised manuscript. So, you will see that I have repeated some of the requests below. The offer of publication is subject to all editorial requests being satisfied. If you have any questions, please do let us know using the contact details provided earlier in this email.

Editorial requests:

- Data availability: Could we ask if you could provide different contact details for data access (i.e. not for a single author) please? When possible, we recommend authors deposit restricted data to a repository that allows for controlled data access. If this is not possible, directing data requests to a non-author institutional point of contact, such as a data access or ethics committee, helps guarantee long term stability and availability of data. Providing interested researchers with a durable point of contact ensures data will be accessible even if an author changes email addresses, institutions, or becomes unavailable to answer requests. Further information is available here: https://journals.plos.org/plosmedicine/s/data-availability#loc-faqs-for-data-policy

- Please address the following that was requested at major revision based on the study’s design:

OBSERVATIONAL STUDIES

* Abstract: Please include the study design, population and setting, number of participants, years during which the study took place (enrollment and follow up), length of follow up, and main outcome measures.

* Please ensure that the study is reported according to the STROBE (or appropriate STROBE extension) guideline (available from: https://www.equator-network.org/reporting-guidelines/strobe) and include the completed STROBE (or STROBE extension) checklist as Supporting Information. Please add the following statement, or similar, to the Methods: "This study is reported as per the Strengthening the Reporting of Observational Studies in Epidemiology (STROBE) guideline (S1 Checklist)." When completing the checklist, please use section and paragraph numbers, rather than page numbers.

* For all observational studies, in the manuscript text, please indicate: (1) the specific hypotheses you intended to test, (2) the analytical methods by which you planned to test them, (3) the analyses you actually performed, and (4) when reported analyses differ from those that were planned, transparent explanations for differences that affect the reliability of the study's results. If a reported analysis was performed based on an interesting but unanticipated pattern in the data, please be clear that the analysis was data driven.

* Please state in the Methods section whether the study had a prospective protocol or analysis plan. If a prospective analysis plan (from your funding proposal, IRB or other ethics committee submission, study protocol, or other planning document written before analyzing the data) was used in designing the study, please include the relevant document(s) with your revised manuscript as a Supporting Information file to be published alongside your study and cite it in the Methods section. A legend for this file should be included at the end of your manuscript. If no such document exists, please make sure that the Methods section transparently describes when analyses were planned, and when/why any data-driven changes to analyses took place. Changes in the analysis, including those made in response to peer review comments, should be identified as such in the Methods section of the paper, with rationale.

- Can you also clarify whether the authors had contact with the patients or not, please? Forgive me for my confusion – it seems like there might have been no contact in some instances but at the same time someone must have contacted the participants for the purpose of being involved in this study? Can you also verify that appropriate consent processes were in place and/or elaborate (both in your rebuttal and in the manuscript) on this please?

- We note your request for clarification as to whether to add further results to the Abstract (for which you received an error from the system). Thank you for your efforts here. As there is a limit to the word count associated with Abstracts, this will have to be kept off the Abstract. 

- In the Abstract, last line of the Methods and Findings section, please expand on the abbreviation ‘APOEe4’.

- In the Author Summary, please expand on the first use of the abbreviate ‘HT’. 

- In the Author Summary, please also address this point initially raised at major revision: ‘Please revise your ‘What do these findings mean?’ section to comply with the following PLOS guidance: “Authors should reflect on the new knowledge generated by the research and the implications for practice, research, policy, or public health. Authors should also consider how the interpretation of the study’s findings may be affected by the study limitations. In the final bullet point of ‘What Do These Findings Mean?’, please describe the main limitations of the study in non-technical language.” ’ 

- Please cite the reference numbers in square brackets. Citations should precede punctuation.

---

## [Editor Report · Decision Letter 3]

16 Oct 2024

Dear Dr Gleason, 

On behalf of my colleagues and the Academic Editor, Carol Brayne, I am very pleased to inform you that we have agreed to publish your manuscript "Long-term cognitive effects of menopausal hormone therapy: Findings from the KEEPS Continuation Study" (PMEDICINE-D-24-02037R3) in PLOS Medicine. Thank you for all the time and effort you and your colleagues put into revising your manuscript.

PRESS

Sincerely, 

Syba

Syba Sunny MBBS, MRes, FRCPath

Associate Editor 

PLOS Medicine

ssunny@plos.org